# A Neglected Extensor Hallucis Longus Tendon Rupture Caused by Arthritic Adhesion

**DOI:** 10.3390/medicina59061069

**Published:** 2023-06-01

**Authors:** Sung Hun Won, Sung Hwan Kim, Young Koo Lee, Dong-Il Chun, Byung-Ryul Lee, Woo-Jong Kim

**Affiliations:** 1Department of Orthopaedic Surgery, Soonchunhyang University Hospital Seoul, 59, Daesagwan-ro, Yongsan-gu, Seoul 04401, Republic of Korea; orthowon@gmail.com (S.H.W.); orthochun@gmail.com (D.-I.C.); 2Department of Orthopaedic Surgery, Soonchunhyang University Hospital Bucheon, 170, Jomaru-ro, Wonmi-gu, Gyeonggi-do, Bucheon-si 14584, Republic of Korea; shk9528@naver.com (S.H.K.); brain0808@hanmail.net (Y.K.L.); 3Department of Orthopaedic Surgery, Soonchunhyang University Hospital Cheonan, 31, Sooncheonhyang 6-gil, Dongnam-gu, Cheonan 31151, Republic of Korea; 129027@schmc.ac.kr

**Keywords:** extensor hallucis longus, contracture, checkrein, arthritis

## Abstract

Extensor hallucis longus tendon injury is relatively rare and is principally caused by a laceration when a sharp object is dropped on the instep. Primary suturing is possible if the injury is acute, but if the tear is chronic, tendon contracture causes the space between the edges of the tear to widen, disrupting the end-to-end connection. In particular, a claw toe or checkrein foot deformity may develop over time due to adhesion of the lower leg tendons near the fracture site or scar. We report on a 44-year-old man who visited our outpatient clinic complaining of pain in the right foot and a hindered ability to extend his great toe. He had enjoyed playing soccer during his schooldays; since that time, the extension of that toe had become somewhat difficult. T2-weighted sagittal magnetic resonance imaging revealed that the continuity of the extensor hallucis longus tendon had been lost at the distal phalangeal base attachment site, and that the region of the proximal tendon was retracted to level of the middle shaft of the proximal phalanx. The findings allowed us to diagnose extensor hallucis longus tendon rupture accompanying osteoarthritic changes in the joint and soft tissues. We performed surgical tenorrhaphy and adhesiolysis. This is a rare case of extensor hallucis longus tendon rupture caused by minor trauma. Arthritis that developed at a young age caused the adhesions. If patients with foot and ankle arthritis show tendon adhesion at the arthritic site, tendon rupture can develop even after minor trauma or intense stretching.

## 1. Introduction

Injury to the extensor hallucis longus (EHL) tendon is relatively rare; the primary cause is a laceration when a sharp object falls in the instep [1,2]. Surgery is required to eliminate the risks of morphological impairment and gait disturbance [3,4]. Primary suturing is possible if the injury is acute, but if the tear is chronic, tendon contracture causes the space between the edges of the tear to widen, disrupting the end-to-end connection [5]. Despite the efforts of several researchers, the overall incidence of EHL rupture has not been established. Anzel et al. studied 1014 cases of various muscle and tendon injuries in the human body and reported that 16 cases (1.5%) were injuries to the extensor tendons in the toe [6]. In the study, open laceration of the foot was most commonly described as the injury mechanism for EHL tendon injuries. However, close traumatic rupture, degenerative rupture, and iatrogenic injury in surgical procedures have also been reported [7]. In particular, EHL tendon rupture tends to be more frequent in martial arts athletes than in the general population [8]. In the case of an acute EHL tendon rupture, surgical treatment with direct end-to-end primary repair of the EHL tendon is recommended. These patients tend to yield favorable outcomes with the self-restoring abilities, and return to activity levels about same as before the injury [9]. A conservative treatment option may be chosen in some cases, like patients with limited activity levels before the injury, patients with too many medical comorbidities to perform operative intervention or patients with ruptures distal to the extensor expansion [10].

Checkrein deformity can be caused by contractures triggered by ischemic changes in muscles and tendons, adhesions to surrounding tissues, tendon insertion into the fracture site, and callus formation after a fracture [11,12]. In particular, a foot claw toe or checkrein deformity can develop over time, attributable to adhesion of the lower leg tendons near the fracture site or scar or to asymptomatic compartment syndrome [13,14]. A checkrein deformity in the flexor hallucis longus (FHL) tendon is more common in clinical practice; there are several reports. Since Clawson reported the first claw toe case after fracture of the tibia in 1974, a few cases of checkrein deformity have been reported [15]. Leitschuh et al. reported a dysfunction of the first toe flexion which was caused by entrapment of the FHL tendon following a fibula fracture [16]. Carr noted an FHL tendon entrapment following a calcaneus fracture, and computerized tomography confirmed the tendon entrapment between fracture fragments [17].

Reverse checkrein deformities, and thus contracture or adhesions of the EHL tendon, are rarer [18,19]. Post-traumatic EHL tendon checkrein deformity can be caused by ischemic changes in the muscles and tendons and adhesions to surrounding tissues. Involuntary extension of the talus during plantarflexion at the ankle joint is due to tendon insufficiency. The tendon dysfunction results in involuntary extension of the first toe during plantarflexion of the ankle joint. Additionally, it also causes muscle weakness of the EHL tendon and limitation in the first toe extension during dorsiflexion of the ankle joint at a neutral position [19]. In Japan, the term reverse checkrein deformity has also been reported, but it is not a commonly used term internationally.

We report a case for which we corrected a chronic reverse checkrein deformity of the EHL tendon when treating an acute, traumatic EHL rupture attributable to adhesions caused by chronic tarsometatarsal (TMT) joint osteoarthritis of a long duration.

## 2. Case Presentation

### 2.1. Preoperative Evaluation

A 44-year-old man visited the outpatient clinic complaining of pain in the right foot and difficulty extending his great toe that had developed 2 weeks earlier when he was playing soccer. His great toe had made a popping sound when it got stuck in the sand. He said that he felt that it was difficult to extend that toe even when playing soccer in his schooldays. After reaching adulthood, he visited the hospital and was told that chronic TMT osteoarthritis explained the problem. However, after the recent trauma, extension had become even more difficult, and the patient requested treatment.

On physical examination, no abrasions or lacerations were found on the right foot. Dimpling was palpable around the proximal interphalangeal joint of the right great toe. He complained of mild to moderate painful swelling around that toe. The toe was plantar-flexed, and the interphalangeal joint extension strength was Grade III. Passive extension was intact, with no limitation of motion. There was no sensory dysfunction anywhere on the foot. There were no biomechanical dysfunctions of any other ligaments and no deformities in any other phalanx or any foot joint except that associated with the EHL tendon.

In plain radiographic evaluations before surgery, compared to the contralateral side, joint space narrowing and bony sclerosis suggested TMT joint osteoarthritis was evident (Figure 1). T2-weighted sagittal magnetic resonance imaging (MRI) revealed that EHL tendon continuity had been lost at the site of attachment to the distal phalangeal base, and the proximal part of the tendon was retracted to the mid-shaft level of the proximal phalanx. Osteoarthritis and a subchondral bone cyst were apparent in the TMT joint of the great toe. The diffuse and heterogeneously high signal intensity around the EHL tendon suggested soft tissue adhesion (Figure 2). MRI revealed nothing further of note in the ligaments, muscles, or bones. Thus, we diagnosed a rupture of the EHL tendon with osteoarthritic changes in the joint and soft tissues, and planned tenorrhaphy and adhesiolysis.

### 2.2. Surgical Procedure

Under spinal anesthesia, the patient was placed supine and a pneumatic tourniquet was applied and pressurized to 300 mmHg. Then, a dorsal Z-shaped incision was created along the EHL from the first TMT joint to the first distal interphalangeal joint. The operative field revealed a total rupture of the EHL tendon at a point 1 cm distal to that joint. Additionally, joint space narrowing with osteophytes was apparent in the first TMT joint, as was severe soft tissue adhesion (Figure 3). The osteophytes of the TMT joint were removed and adhesiolysis was performed to treat severe adhesions (Figure 4). Next, the EHL tendon was core-sutured using the locking method and augmented. Finally, after the wound was sutured, a short leg splint was applied with the ankle in maximal dorsiflexion and the great toe was extended.

### 2.3. Postoperative Care

For the first 2 weeks after surgery, the distal interpharyngeal and ankle joints were immobilized using a non-weight-bearing short leg splint (below the knee) in the neutral position to protect the wound and the repaired tendon. Then, as a component of rehabilitation, passive extension of the distal interpharyngeal joint via manual manipulation was commenced to prevent joint contracture. Four weeks after surgery, active extension of the toe and partial weight-bearing of the foot (with boots) began. Six weeks after surgery, the brace was removed and full weight-bearing began. A night splint was utilized until five months, jogging was allowed at seven months, and release to full sports was achieved at nine months post-operation.

At the 3 month follow-up, active hallux extension was possible over the full range of motion. The patient had no symptoms and reported no discomfort in daily activities. One year after surgery, the hallux extension strength was very close to that of the contralateral side, and tendon contraction was evident under the skin. The AOFAS (American Orthopaedic Foot and Ankle Society) Hallux MP-Interphalangeal Scale score improved from 73 to 90 and the Foot and Ankle Ability Measure scores ranged from 64% to 94% (ADL subscale) and 72% to 94% (sports subscale).

## 3. Discussion

The EHL tendon arises from about the center of the anterior surface of the fibula and also from the interosseous membrane of the leg. The muscle belly lies between the extensor digitorum longus and the tibialis anterior muscles, and then becomes a tendon that passes under the superior and inferior extensor retinacula to become finally inserted into the dorsobasal aspect of the distal phalanx of the hallux [20]. The EHL is supplied by the anterior tibial artery and innervated by the deep peroneal nerve [21]. The EHL is responsible for hallux extension, ankle dorsiflexion, and foot inversion. Thus, its rupture or adhesion disables these functions. In the diagnostic process, the first step is to perform a clinical examination. In the clinical examination, an EHL tendon rupture patient will often complain of a limitation in extension strength or range of motion in the IP joint of the first toe and associated deformity and dysfunction of flexion strength. Foot plain radiographs should be taken routinely to exclude the possibility of an avulsion fracture in the tendon insertion site. Further evaluation with ultrasound or MRI might be used to confirm the tendon rupture by finding a loss in tendon continuity, measuring the size of the gap between the proximal part and distal part of tendon, evaluating the quality of the remaining tissue, and also evaluating the possible concomitant injuries in the surrounding bone and tissue structure [7].

It seems that there is a consensus on acute EHL tendon rupture treatments with direct primary repair. However, it is controversial which option is the best method for nonrepairable EHL tendon injury treatment [7]. Indication for when to treat EHL tendon rupture with EHL reconstruction has not been given. When possible, treatment with direct primary repair of the acute EHL tendon rupture results in a good prognosis [22]. In case of a chronic rupture which is longer than six weeks, the EHL tendon may be retracted far away and there might be scar tissue [23]. Specifically, patients with experience of previous failed primary repair or preceding symptoms before tendon rupture may require more tendon debridement [24]. These patient factors may mean that the patient’s injury has a large tendon gap that cannot be directly repaired and needs additional procedures or a more complicated surgical option.

Wrist arthritis can trigger extensor tendon ruptures that are not always clinically apparent, especially when the peritendinous synovium adheres to the tendon course [25]. Extensor tendon rupture associated with osteoarthritis of the distal radioulnar joint is very rare compared to spontaneous extensor tendon rupture in patients with rheumatoid arthritis [26]. Patients usually tolerate tendon pathologies for a long time because they develop slowly. After tendon rupture, some “artificial” function may remain, attributable to an intact peritendineum. Spontaneous rupture of a damaged extensor tendon reflects inflammatory or reactive involvement of the tendon in a wrist disease [25]. Thus, degenerative changes in the distal radioulnar joint increase the risk of extensor tendon rupture [26]. Likewise, in the foot and ankle, osteoarthritis and arthritis can cause neglected rupture of an extensor tendon such as the EHL. Arthritis triggers adhesions along the course of the tendon, reducing mobility and the power of the tendon. Thus, a weakened tendon may be ruptured by a trauma so minor that the patient cannot remember it. Therefore, we recommend that patients with arthritis of the foot or ankle joints based on history or tests should engage in tendon stretching or rehabilitation to prevent rupture by minor trauma or to strengthen an extensor tendon weakened by a neglected tendon rupture.

The limitation of our study is that we report on only one patient. We cannot offer a general treatment method with only this case. Further multi-institutional research with more patients is required to verify the association between foot and ankle joint osteoarthritis and neglected EHL tendon ruptures.

## 4. Conclusions

We describe a rare case of a primary repair of a neglected EHL tendon rupture caused by arthritic adhesion. The functional results were comparable to those of the contralateral side. It should be noted that in patients with limitations in range of motion attributable to osteoarthritis, forcible movement in the opposite direction may trigger ligament rupture. In patients with foot and ankle arthritis and tendon adhesion at the arthritic site, tendon rupture can occur even after minor trauma or intense stretching.

## Figures and Tables

**Figure 1 medicina-59-01069-f001:**
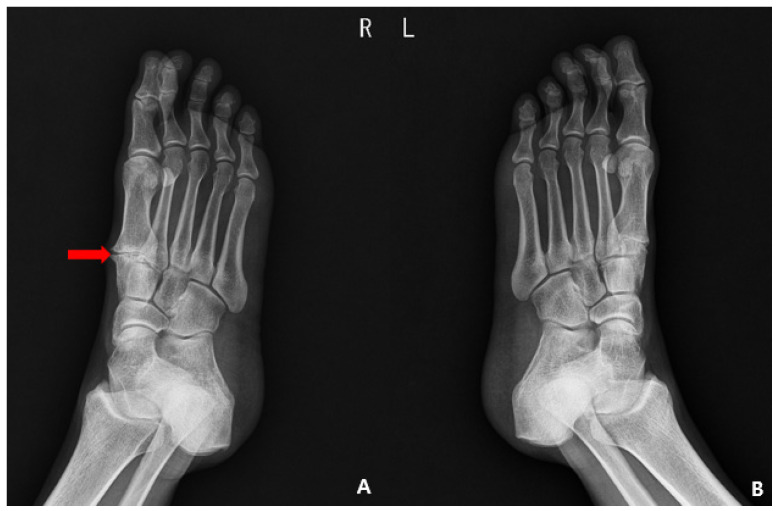
Preoperative plain radiographs showing osteoarthritis with joint space narrowing, sclerotic changes, and osteophytes in the first TMT (Tarsometatarsal) joint (arrow). (**A**) Oblique view of the site of injury, (**B**) oblique view of the contralateral side.

**Figure 2 medicina-59-01069-f002:**
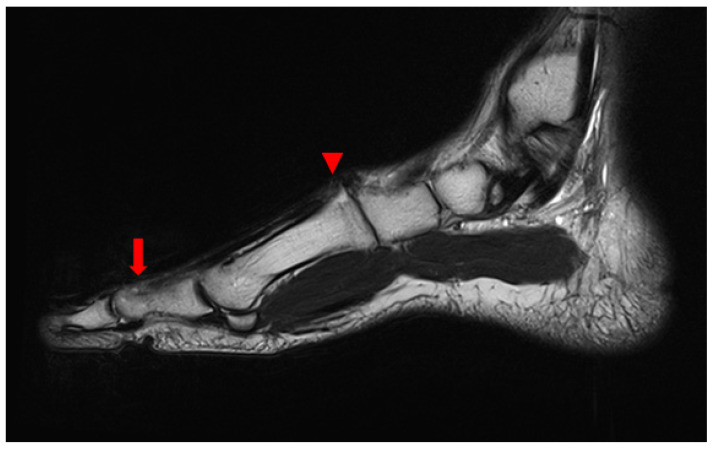
Preoperative T2-weighted sagittal magnetic resonance image that revealed that the EHL was discontinuous. Diffuse and heterogeneously high signal intensity was apparent in soft tissue around the EHL (Extensor hallucis longus) tendon in the proximal region of the proximal phalanx (arrow) and the TMT joint (arrowhead).

**Figure 3 medicina-59-01069-f003:**
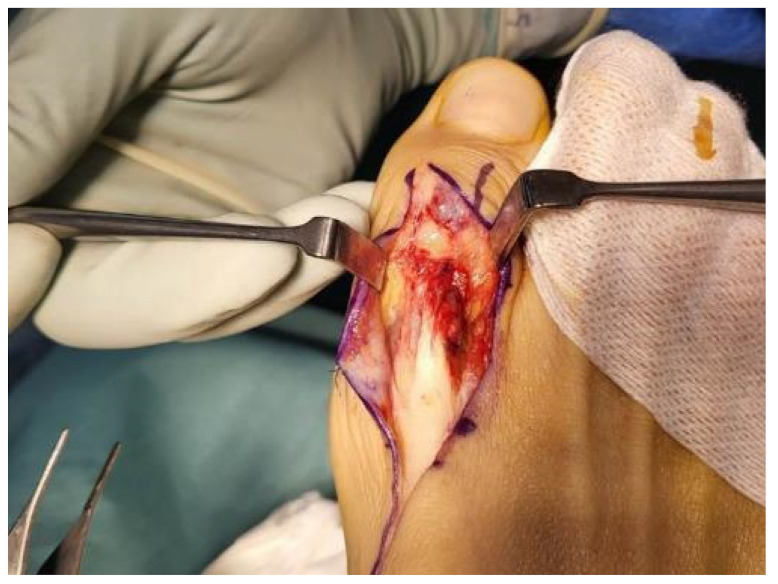
The EHL tendon was severed at a point 1 cm distal to the interpharangeal joint. Severe soft tissue adhesion along the EHL tendon was observed.

**Figure 4 medicina-59-01069-f004:**
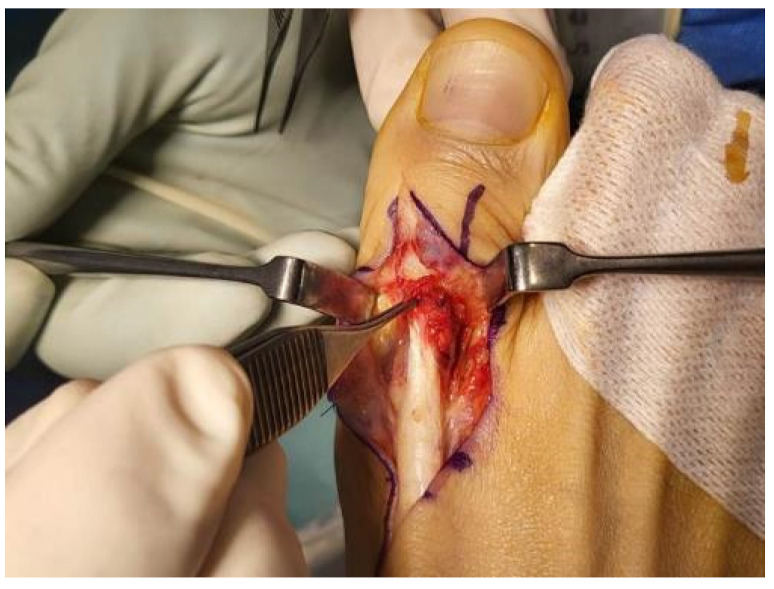
The osteophytes were removed and adhesiolysis was performed using Metzembaum to treat severe adhesions.

## Data Availability

Data sharing is not applicable to this article because no datasets were made or analyzed during this study.

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
