# Peer review of "A Neglected Extensor Hallucis Longus Tendon Rupture Caused by Arthritic Adhesion"

_medicina, 2023, doi:10.3390/medicina59061069_

Round 1

Reviewer 1 Report

The aim of the reviewed study is the presentation of a case of surgical repair of a neglected extensor hallucis longus tendon rupture caused by arthritis adhesion. The condition is rare and rarely described (only 15 published articles about extensor hallucis rupture with no mention of arthritis as a cause in PubMed database from the last 10 years), so the topic addresses the gap in the orthopedics field. The presented surgical technique and postoperative care can be very helpful for doctors who will encounter this problem in the future. The paper is well-constructed and clear. The discussion was carried out correctly. The presented conclusions are consistent. The figures are well chosen and described. References typical and actual.  

Shortcomings are listed below:

-       Please add  AOFAS and ADL to the abbreviations section – line 194.

-       There should be dots instead of commas in line 199 (Author Contributions section).

-       Check the format of the references. It is inconsistent and does not fully comply with the requirements of the publishing house.

Minor editing of English language required. Single punctuation and spelling errors.

Author Response

Dear reviewer,

We really appreciate to your kind review comments.

Actually, those comments are really helpful to our article to be more completed.

Below are our answer to your comments and questions.

-       Please add  AOFAS and ADL to the abbreviations section – line 194.

-       There should be dots instead of commas in line 199 (Author Contributions section).

-> Thank you for kind advise. We correct those mistake

-       Check the format of the references. It is inconsistent and does not fully comply with the requirements of the publishing house.

-> We feel very thankful and sorry about those mistake. We correct reference format upon your advise.

King regards,

Reviewer 2 Report

Thank you for the opportunity to review a very interesting article entitled: „A neglected extensor hallucis longus tendon rupture caused by arthritic adhesion”.

The manuscript is well written, however I have some minor concerns to the work:

- The illustrations attached in the work are clear and well described.

- The work could include measurements of anatomical structures to increase the scientific value of the work, but the current form is the most correct.

However, I also have minor concerns about linguistic proofreading. I have included my corrections in the attached document.

I have no major concerns to this work. 

I have some minor concerns about linguistic proofreading. I have included my corrections in the attached document.

Author Response

Dear review,

We really appreciate to your kind review comments.

Actually, those comments are really helpful to our article to be more completed.

We specially feel thankful about your kind linguistic correction

King regards,